# Effects of Polyacrylamide, Biochar, and Palm Fiber on Soil Erosion at the Early Stage of Vegetation Concrete Slope Construction

**Lu Xia** [1], **Bingqin Zhao** [1], **Ting Luo** [1], **Yakun Xu** [1], **Shiwei Guo** [1], **Wennian Xu** [1] **and Dong Xia** [1,2,*]

1   Hubei Provincial Engineering Research Center of Slope Habitat Construction Technique Using Cement-Based Materials, Three Gorges University, Yichang 443002, China
2   Key Laboratory of Urban Land Resources Monitoring and Simulation, Ministry of Natural Resources, Shenzhen 518034, China
*   Correspondence: xiadongsanxia@163.com; Tel.: +86-13487219472

**Abstract:** The goal of this research is to investigate strategies to increase the erosion resistance of the slope surface during the early stages of vegetation concrete construction, as well as to offer a scientific foundation for improving vegetation concrete formulation. Simulated rainfall experiments were carried out at 2 different slope gradients (50° and 60°), 2 different rainfall intensities (60 and 120 mm·h$^{-1}$), and 4 treatments (CK-no additive, 0.4% P-polyacrylamide, 4% C-biochar, and 0.4% F-palm fiber). PAM, palm fiber, and biochar significantly reduced the initial runoff time of the vegetation concrete slope by an average of 47.03%, 46.41%, and 22.67%, respectively ($p < 0.05$). The runoff rate of each slope under different conditions increased with the expansion of rainfall duration and then fluctuated and stabilized, whereas the erosion rate decreased and then fluctuated and stabilized. PAM and palm fiber both increased runoff rates while decreasing erosion rates, but biochar increased both runoff rates and erosion rates. The runoff reduction benefits of PAM, palm fiber, and biochar were −69.84~−1.97%, −68.82~−14.28% and −63.70~−6.80%, respectively, while the sediment reduction benefits were 69.21~94.07%, −96.81~−50.35%, and 36.20~60.47%, respectively. PAM and palm fiber both have obvious sediment reduction benefits and can be used in the ecological restoration of high and steep slopes in areas with heavy rainfall.

**Keywords:** biochar; palm fiber; PAM; runoff and sediment reduction benefits; vegetation concrete

## 1. Introduction

The development of large-scale infrastructure such as highways, railways, and hydropower stations has resulted in significant soil erosion and the formation of numerous exposed slopes [1–3]. The habitats of these exposed slopes are exceedingly inadequate, and it is difficult to restore them to the original ecological level only by the power of natural forces. In response to this phenomenon, scholars and engineering professionals around the world have developed a series of slope vegetation restoration technologies to speed up the ecological restoration of slopes with the help of human power, such as Vegetation Concrete Base Spraying (CBS) [4], Thick Layer Base Material Spraying (TBS) [5], External-Soil Spray Seeding (ESSS) [6], and Frame Beam + External-Soil (FBES), etc. CBS is an ecological restoration technology that mixes soil, cement, organic materials, amendment of habitat material and greening seeds in proportion to the characteristics of slopes, climatic environment, and plant needs to form a specific ecological substrate formula to restore different types of slopes in different environments. It has been applied in ecological restoration projects of slopes in more than 20 provinces and cities in China and has become an industrial standard due to its advantages including high strength of substrate, close combination with the slope surface, and better ecological restoration effect [3,4]. CBS has achieved significant

ecological benefits and is the primary technology for rapid vegetation restoration of bare slopes, especially high and steep hard slopes in China [3].

Many scholars have conducted extensive studies on CBS with fruitful results over the last two decades. These studies primarily focus on the ratio of vegetated concrete substrate, physical and mechanical properties, biological nutrient characteristics, and plant physiological growth, but there is very little research on the erosion of vegetation concrete slopes, especially at the early stages of engineering construction [7–11]. The curing capacity of cement in vegetation concrete substrates does not completely operate at this early stage of construction, and the slope surface is still vulnerable to rainwater erosion. Xia et al. [12] made vegetation concrete test blocks with varying cement content and curing ages to conduct flume scouring tests, revealing that the erosion resistance of vegetation concrete substrate increases with increasing cement content and curing age in a certain range. Li et al. [13] and Zhou et al. [14] used self-made slope erosion models to carry out anti-erosion tests on vegetation concrete slopes during the early stages of construction. The slope erosion was discovered to be mostly layered surface erosion, with a clear scale phenomenon. The amount of sediment erosion increased as the slope rate and rainfall intensity increased. The vegetation concrete slope has certain anti-erosion abilities at the early stage of construction, and there is still room to improve the anti-erosion ability of vegetation concrete substrate under heavy rainfall circumstances. Substrate creates necessary growth conditions for plants on the slope and is the basis for stable growth of vegetation [15]. Concurrent with runoff and erosion, there are the losses of the nutrients [16,17], which seriously affect the ecological restoration of exposed slopes.

Biochar and polyacrylamide (PAM) are two commonly utilized soil structure conditioners in the field of soil and water conservation [18–20]. Biochar is a highly aromatized solid product formed by high-temperature cracking of various plant biomass in aerobic or anaerobic environments, with well-developed pores, a large specific surface area, and a high capacity for ion adsorption [21–23]. When applied to soil, biochar can improve soil physical and chemical characteristics, optimize soil structure, increase soil water-holding capacity, and improve fertility retention [24–29]. PAM is a water-soluble linear polymer with the advantages of excellent water absorption and retention, recurrent water absorption, and gradual release, which all benefit soil water infiltration, water retention, and plant water utilization [30–33]. At the same time, PAM can also promote the formation of large agglomerates by adsorbing soil particles through group bonding in the molecular structure [34]. Fiber reinforcement technology is gaining popularity in the field of soil improvement because it can greatly enhance mechanical properties such as shear strength, tensile strength, and bearing capacity, as well as increase soil strength [8,35,36]. All three can reduce the occurrence of soil erosion by modifying the soil structure and enhancing the physical and mechanical properties of the soil [37–39].

In view of the positive effects of the three materials in soil improvement, this paper simulates the vegetation concrete slope at the early stage of construction using a self-designed slope scour model with three materials (polyacrylamide, biochar, and palm fiber) as external admixtures and conducts indoor artificial rainfall experiments using the vegetation concrete slope without external admixtures as the control. The effects of different rainfall intensities and slopes on the runoff and sediment production process of the vegetation concrete slope as well as the benefits of runoff and sediment reduction of the base material with external admixture were compared and analyzed, which can provide data and support for improving the formulation of the vegetation concrete base material and its application in areas with heavy rainfall.

## 2. Materials and Methods

### 2.1. Materials for Experiments

The materials required for the experiments were mainly planting soil, cement, organic materials, organic fertilizer, amendment of habitat material, palm fiber, polyacrylamide, biochar, and water. The planting soil was collected from the common yellow-brown soil

in the suburbs of Yichang City, Hubei Province. After collection and transportation, it was sun-dried and crushed before being passed through a 5 mm sieve for future usage. The dry bulk weight of the planting soil was 1.24 g/cm$^3$, the organic matter content was 1.5%, and the particle composition was 20.8%, 58.6%, and 20.6% for sand, slit, and clay grains, respectively. Ordinary silicate cement (P.O42.5) was used. The organic material is poplar sawdust from the Yichang City Lumber Plant. The amendment of habitat material is a patented product of the Hubei Runzhi Ecological Technology Company. The organic fertilizer is produced by the Zhubang Biological Company. Palm fiber is coconut palm fiber with a moisture content of 12%, a fiber strength of greater than 255 N/D, and a length of around 3 cm. PAM is anionic, its molecular weight is 15 million, and the hydrolysis degree is between 10% and 30%. The particle size of biochar is about 200 mesh, and the pH is about 8.0. The water is regular tap water.

### 2.2. Experimental Layout

The experimental equipment uses a self-designed slope test model, which consists mostly of a metal box with an adjustable slope, a diversion channel, a support rod, and a lifting device (Figure 1a,b). The lifting device is used to adjust the slope of the box, which has a range of 40~70°. After adjusting the slope, the support bar is supported at the bottom of the box to fix the box. The inside measurements of the box are 150 cm × 67 cm × 20 cm (length, width, and height). During the test, the diversion channel is linked to the lower half of the box, and a portion of it is immersed in the box to guarantee that all the scouring sediment and runoff is routed through the diversion channel. The experiment was carried out with the BX-1 combined side-jet rainfall instrument developed by the Institute of Soil and Water Conservation, Chinese Academy of Sciences. The parameters are as follows: the maximum rainfall height is 7 m, the effective rainfall area is 5 m × 7 m, and rainfall uniformity exceeds 80% (Figure 1c). Artificial simulated rainfall experiments were carried out with 2 slopes (50° and 60°), 2 rainfall intensities (60 and 120 mm·h$^{-1}$), and 4 treatments (existing substrate without any admixtures (CK), 0.4% PAM (P), 4% biochar (C) and 0.4% palm fiber (F) added to the existing substrate, respectively) to investigate the effects of admixtures on slope erosion at the early stage of vegetation concrete slope construction under different conditions (Table 1). The rainfall lasted 30 min, and each rainfall experiment was repeated 3 times for a total of 48 rainfall episodes.

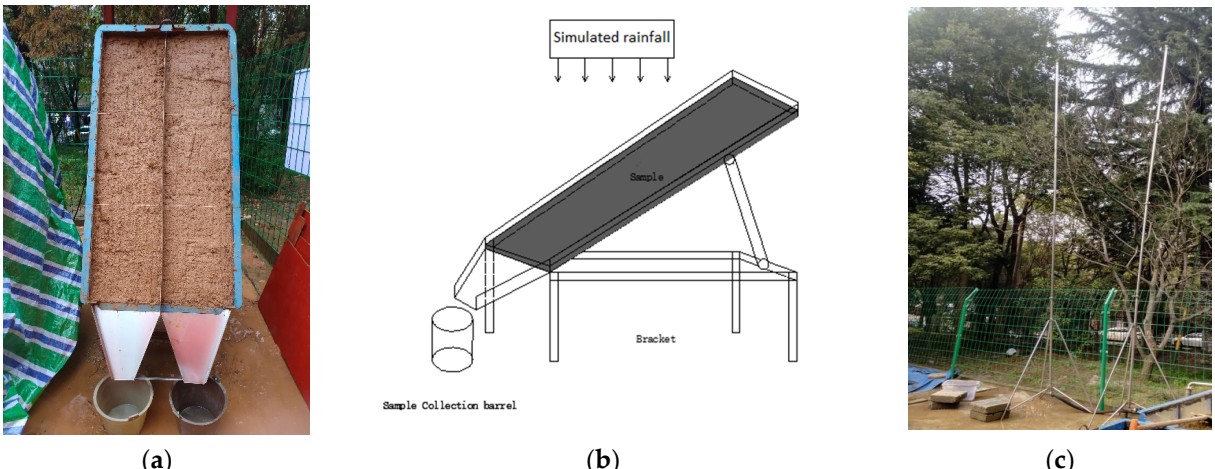

| (a) | (b) | (c) |

**Figure 1.** Schematic diagram of the test device and rainfall instrument. (**a**) runoff plot. (**b**) solid model. (**c**) BX-1 combined side-jet rainfall instrument.

**Table 1.** Proportion of Vegetation Concrete Substrate.

| Substrate | Planting Soil | Cement | Organic Materials | Amendment of Habitat Material | Organic Fertilizer |
|---|---|---|---|---|---|
| foundation course | 100 | 6 | 5 | 3 | 2 |
| surface course | 100 | 3 | 5 | 1.5 | 2 |

Note: The above constituent ratios are mass ratios, and the quality of planted soil is the basis for calculation.

### 2.3. Experimental Procedure

According to engineering practice monitoring, the erosion of vegetation concrete slopes frequently occurs at the surface layer of the substrate. Therefore, the external admixture was only administered to the surface layer of the substrate. Meanwhile, in order to cut costs and simplify the construction process, PAM, biochar, and palm fiber were mixed into the substrate as a dry mix. To construct 4 treatments of substrate, the components were thoroughly mixed according to the specified ratio and amount of external admixture, and water was added multiple times until the moisture content of the substrate reached 20% (the optimal moisture content for engineering applications). A tiny spraying equipment was then used to spray the substrate onto the slope. The spraying of the substrate followed the NB/T 35082-2016 'Technical code for eco-restoration of vegetation concrete on steep slope of hydropower projects' [4]. The bottom layer of 8 cm was sprayed first, followed by the surface layer of 2 cm. After the substrate had been sprayed, the slope surface was suspended and then covered with plastic film. After 24 h of static maintenance, simulated rainfall experiments were performed. After calibrating the rainfall intensity to the desired rainfall intensity, the plastic film was removed, and the timing began. The initial runoff time was recorded, and the timing was restarted once the slope surface generated continuous runoff. The rain stopped 30 min after the runoff began. The runoff samples were continuously collected with plastic buckets. The plastic buckets were replaced every 3 min, and a total of 10 samples were received. After the rain stopped, the runoff samples from each step were weighed and recorded before being placed in a chamber for 24 h to remove the supernatant and separate the sediment samples. The quantity of soil erosion was calculated using the weight of the sediment sample after drying, and the runoff sample besides the dry weight of sediment was runoff.

### 2.4. Method of Data Collection and Analysis

The soil potential of hydrogen was measured using a glass electrode in a 1:2.5 soil/water solution, and organic matter was measured by the potassium dichromate method [40]. The soil bulk density was measured by the cutting ring method, and soil particles were determined by the TopSizer laser particle size analyzer (SCF-108, Zhuhai Oaxac Instrument Company, Zhuhai, China).

The initial runoff time ($T$, s), runoff rate ($R$, L·min$^{-1}$), erosion rate ($E$, g·min$^{-1}$), cumulative runoff ($R_{CR}$, L), cumulative sediment ($E_{CE}$, g), total runoff ($R_T$, L), total sediment ($E_T$, g), runoff reduction benefit ($CR$, %), and sediment reduction benefit ($CE$, %) were used to analyze the characteristics of runoff and sediment yield process on vegetation concrete slopes. The calculation formula for each index is as follows:

$$R = \frac{R_t}{t} \tag{1}$$

$$E = \frac{S_t}{t} \tag{2}$$

$R_t$ is the runoff collected during the sampling time, L. $S_t$ is the sediment content in the runoff collected during the sampling period, g. $t$ is the sampling time (3 min).

$$CR_i = \frac{R_{TCK} - R_{Ti}}{R_{TCK}} \tag{3}$$

$$CE_i = \frac{E_{TCK} - E_{Ti}}{E_{TCK}} \tag{4}$$

$R_{TCk}$ and $R_{Ti}$ ($i$ = P, C, F) are total runoff under CK, P, C, and R, respectively. $E_{TCk}$ and $E_{Ti}$ ($i$ = P, C, F) are total sediment under CK, P, C, and R, respectively.

SPSS21.0 was used to analyze whether there were significant differences between different treatments under the same rainfall intensity and slope ($p < 0.05$). All figures are completed with Origin 17.0.

## 3. Results

### 3.1. Effects of Polyacrylamide, Biochar, and Palm Fiber on Initial Runoff Time of Slope

The initial runoff time of each slope is shown in Figure 2. Except for P and F at 120 mm·h$^{-1}$ and 60°, the runoff time of different slopes differed significantly ($p < 0.05$) under the same rainfall intensity and slope conditions. The initial runoff time of the P, C, and F slopes was smaller than that of the CK slopes, indicating that the addition of PAM, palm fiber, and biochar shortened the initial runoff time of vegetation concrete slopes, and the three kinds of external admixtures had different effects on the initial runoff time of slopes. In general, as compared with the CK slopes, the initial runoff under the P, C, and F slopes was shortened by 40.24~53.50%, 32.50~58.15%, and 15.78~35.35%, respectively, with an average of 47.03%, 46.41%, and 22.67%, respectively. With the increase in rainfall intensity and slope gradient, the initial runoff time of different slopes also decreased.

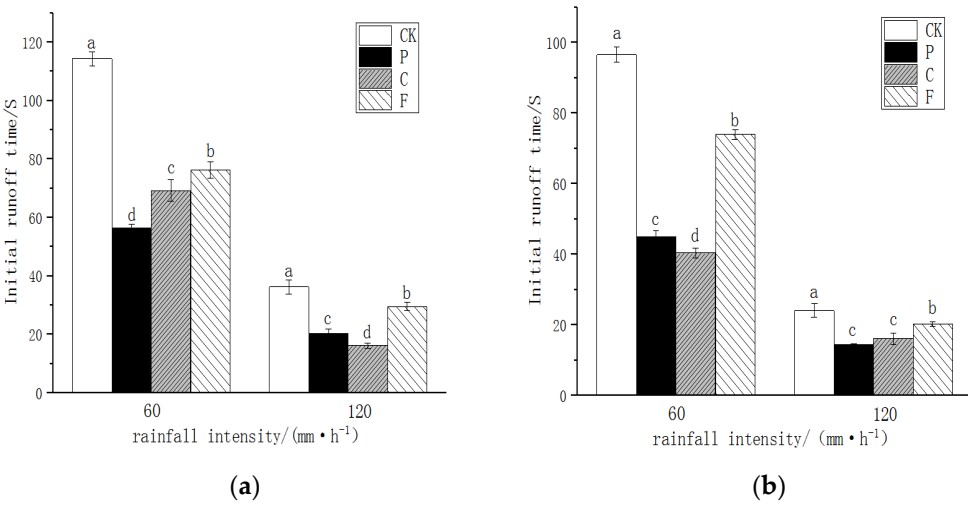

**Figure 2.** Initial runoff time of the four treatments under different rainfall intensity and slope gradient. (**a**) Slope gradient 50°. (**b**) Slope gradient 60°. Note: CK-Control check; P-0.4% PAM; C-4% biochar; F-0.4% palm fiber; Different letters under the same rainfall intensity and slope gradient indicate significant differences between different treatments ($p < 0.05$).

### 3.2. Effect of Polyacrylamide, Biochar, and Palm Fiber on Runoff and Sediment Process of Slope

The runoff rate of each slope increased first and then fluctuated steadily with the extension of rainfall duration, and the runoff tended to be stable after 12~24 min of rainfall duration under different rainfall intensities and slope gradients (Figure 3). In general, the runoff rate under the P, C, and F slopes was generally greater than that of the CK slope in each runoff period, although the order of runoff rate under the P, C, and F slopes varied depending on rainfall intensity and slope gradient. The change curve of runoff rate under the P, C, and F slopes was substantially greater than that of the CK slope at 60 mm·h$^{-1}$, 50° and 60 mm·h$^{-1}$, 60°. The change curve of runoff rate under the four treatments was relatively close at 120 mm·h$^{-1}$, 50° and 120 mm·h$^{-1}$, 60°. It is shown that PAM, palm fiber, and biochar increase the runoff of a vegetation concrete slope, with the impact being more pronounced when the rainfall intensity is low.

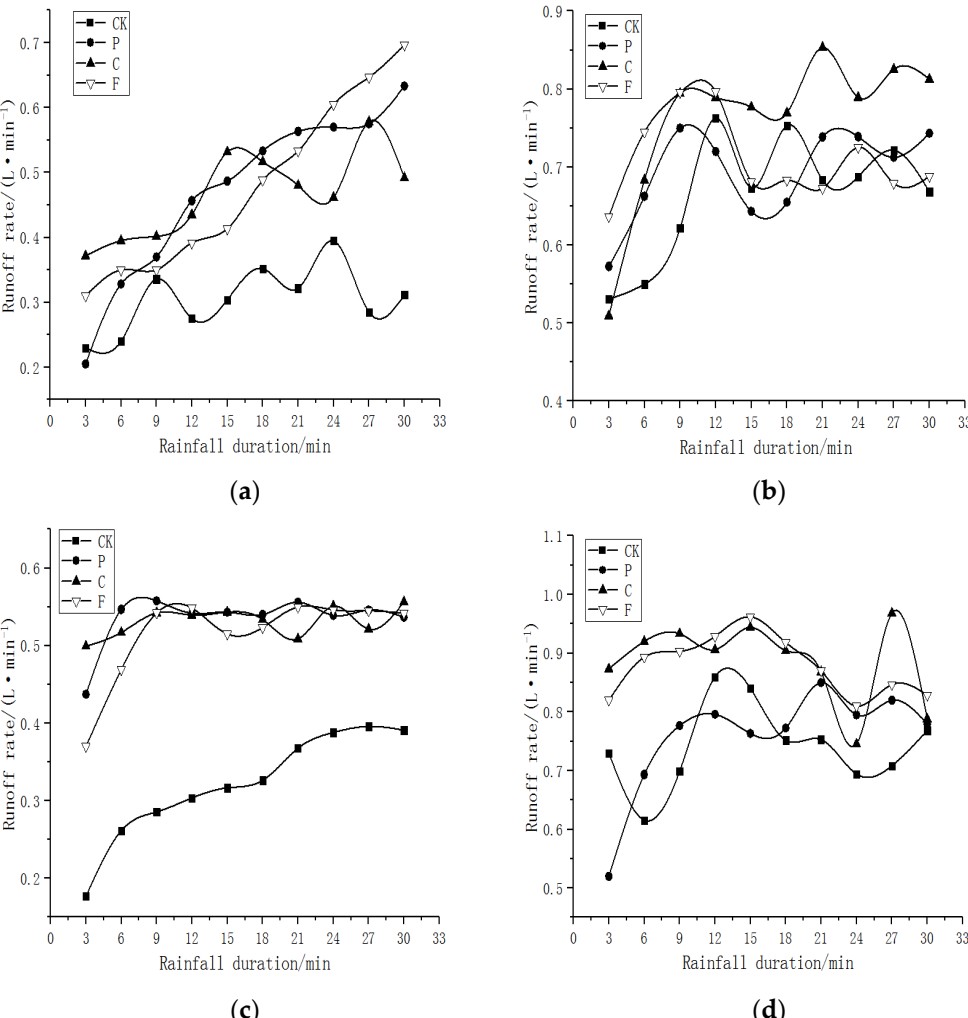

**Figure 3.** Runoff rate of the four treatments under different rainfall intensity and slope gradient. (**a**) 50° and 60 mm·h$^{-1}$. (**b**) 50° and 120 mm·h$^{-1}$. (**c**) 60° and 60 mm·h$^{-1}$. (**d**) 60° and 120 mm·h$^{-1}$.

Unlike the change rule of the slope runoff process with the increasing rainfall duration, the change trend of each slope had a distinct difference (Figure 4). With increasing rainfall duration, the erosion rate of the C slope decreased first and then fluctuated and stabilized, and the decline was significant, but the erosion rate of the P, F, and CK slopes constantly fluctuated and remained steady. Simultaneously, when other parameters were constant, the erosion rate increased with increasing slope gradient and rainfall intensity. According to the analysis of erosion rates under four different slope situations, each runoff period essentially followed the rule of C > CK > F > P. With the passage of time, the erosion rate difference between the C slope and the CK slope became lower and smaller, whereas the erosion rate difference between the F slope, the P slope, and the CK slope remained steady or slowly increased. This has shown that biochar enhanced the erosion rate of vegetation concrete slopes, PAM and palm fiber successfully decreased the erosion rate of vegetation concrete slopes, and PAM and palm fiber always maintained their anti-erosion function while rainfall duration passed.

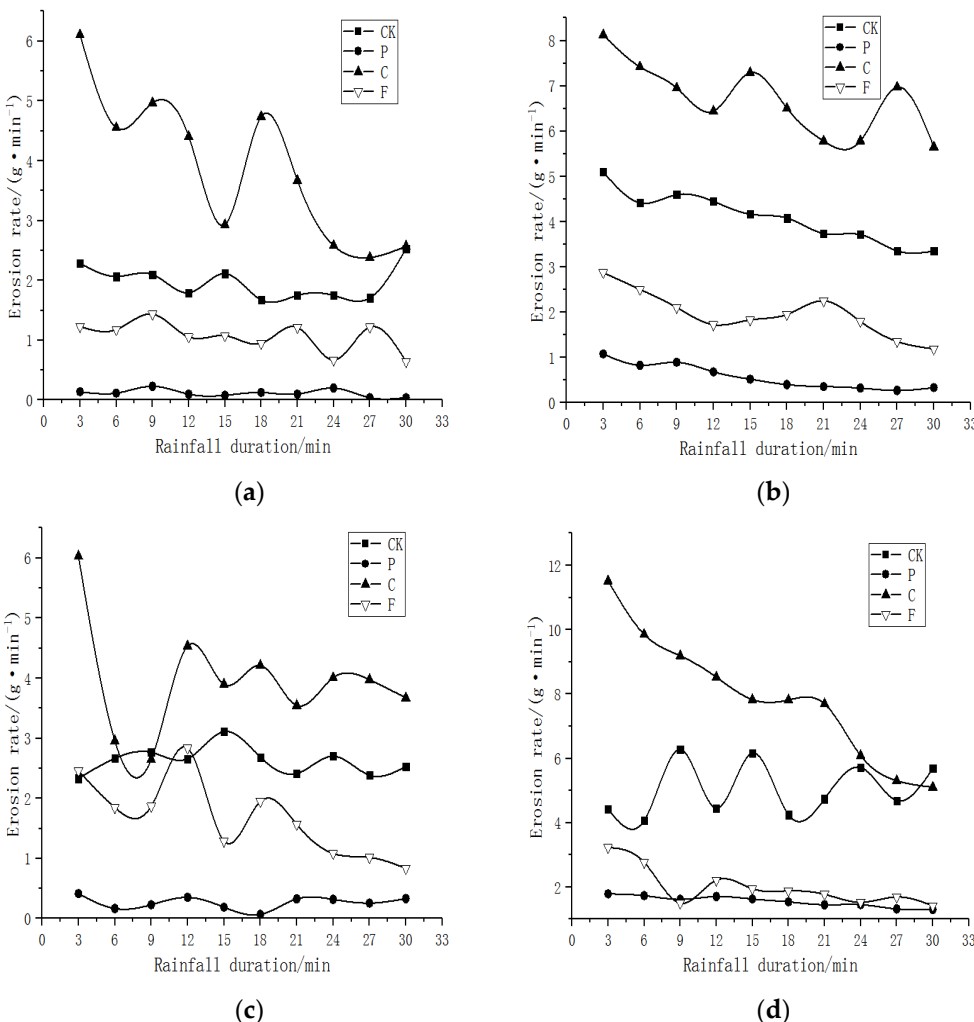

**Figure 4.** Erosion rate of the four treatments under different rainfall intensity and slope gradient. (**a**) 50° and 60 mm·h⁻¹. (**b**) 50° and 120 mm·h⁻¹. (**c**) 60° and 60 mm·h⁻¹. (**d**) 60° and 120 mm·h⁻¹.

### 3.3. Water-Sediment Relationship of the Slope under Different Conditions

The relationship between cumulative runoff and cumulative erosion can be quantitatively reflected by the dynamic system of water and sediment in the slope erosion process. The cumulative erosion and cumulative runoff of each rainfall in the test were fitted with a function, and the linear function relationship between the two was $S_L = AR_L + B$, where $S_L$ was the cumulative erosion, $R_L$ was the cumulative runoff, and the fitting coefficient $R^2$ was greater than 0.95, as shown in Table 3.

**Table 2.** Cumulative runoff and cumulative sediment yield fitting equation.

| Slope | Slope Gradient (°) | Rainfall Intensity (mm·h⁻¹) | | | |
|---|---|---|---|---|---|
| | | **60** | | **120** | |
| CK | 50 | y = 5.882x + 4.770 | R² = 0.995 | y = 5.771x + 11.002 | R² = 0.995 |
| | 60 | y = 7.852x + 6.161 | R² = 0.991 | y = 6.773x + 0.836 | R² = 0.998 |
| P | 50 | y = 0.229x + 0.622 | R² = 0.956 | y = 0.703x + 3.788 | R² = 0.950 |
| | 60 | y = 0.451x + 0.456 | R² = 0.989 | y = 1.941x + 3.757 | R² = 0.996 |
| C | 50 | y = 7.550x + 18.170 | R² = 0.976 | y = 8.197x + 17.493 | R² = 0.997 |
| | 60 | y = 7.136x + 4.616 | R² = 0.998 | y = 8.481x + 20.580 | R² = 0.993 |
| F | 50 | y = 2.090x + 4.681 | R² 0.963 | y = 2.624x + 4.759 | R² = 0.996 |
| | 60 | y = 3.059x + 6.738 | R² 0.975 | y = 2.069x + 6.422 | R² = 0.997 |

Note: x is the cumulative runoff, y is the cumulative erosion.

### 3.4. Water-Sediment Relationship of the Slope under Different Conditions

The relationship between cumulative runoff and cumulative erosion can be quantitatively reflected by the dynamic system of water and sediment in the slope erosion process. The cumulative erosion and cumulative runoff of each rainfall in the test were fitted with a function, and the linear function relationship between the two was $S_L = AR_L + B$, where $S_L$ was the cumulative erosion, $R_L$ was the cumulative runoff, and the fitting coefficient $R^2$ was greater than 0.95, as shown in Table 3.

**Table 3.** Cumulative runoff and cumulative sediment yield fitting equation.

| Slope | Slope Gradient (°) | Rainfall Intensity (mm·h$^{-1}$) | | | |
| --- | --- | --- | --- | --- | --- |
| | | **60** | | **120** | |
| CK | 50 | y = 5.882x + 4.770 | R$^2$ = 0.995 | y = 5.771x + 11.002 | R$^2$ = 0.995 |
| | 60 | y = 7.852x + 6.161 | R$^2$ = 0.991 | y = 6.773x + 0.836 | R$^2$ = 0.998 |
| P | 50 | y = 0.229x + 0.622 | R$^2$ = 0.956 | y = 0.703x + 3.788 | R$^2$ = 0.950 |
| | 60 | y = 0.451x + 0.456 | R$^2$ = 0.989 | y = 1.941x + 3.757 | R$^2$ = 0.996 |
| C | 50 | y = 7.550x + 18.170 | R$^2$ = 0.976 | y = 8.197x + 17.493 | R$^2$ = 0.997 |
| | 60 | y = 7.136x + 4.616 | R$^2$ = 0.998 | y = 8.481x + 20.580 | R$^2$ = 0.993 |
| F | 50 | y = 2.090x + 4.681 | R$^2$ 0.963 | y = 2.624x + 4.759 | R$^2$ = 0.996 |
| | 60 | y = 3.059x + 6.738 | R$^2$ 0.975 | y = 2.069x + 6.422 | R$^2$ = 0.997 |

Note: x is the cumulative runoff, y is the cumulative erosion.

By combining the concept of function formula with the physical meaning of actual runoff and sediment yield, it could be seen that regression parameter $A$ was the indicative parameter in the process of runoff and sediment yield, and parameter $A$ was defined as the coefficient of sediment yield rate. $A > 0$ indicated that the cumulative erosion increased as cumulative runoff increased. When the functional relationship of all rainfall events was summarized and compared, it was found that coefficient $A$ had a specific changing rule. The $A$ value of each slope showed C > CK > F > P under the same slope and rainfall intensity. The $A$ value of biochar, palm fiber, and PAM slope was 0.91~1.42 times, 0.31~0.45 times, and 0.04~0.29 times that of the CK slope. Except for CK at 50° and 60°and F at 60°, the $A$ value of each slope increased with the increasing rainfall intensity. Except for C at 60 mm·h$^{-1}$ and F at 120 mm·h$^{-1}$, the $A$ value of each slope also increased with the increasing slope gradient.

### 3.5. The Influence of Polyacrylamide, Biochar, and Palm Fiber on Runoff and Sediment Reduction of Slope

Table 4 shows the total runoff of each slope as well as the runoff reduction benefits of biochar, palm fiber, and PAM at different rainfall intensities and slope gradients. Except for the P and CK slopes at 120 mm·h$^{-1}$ and 60° conditions, the total runoff of the P, C, and F slopes was considerably greater than that of the CK slopes ($p < 0.05$), showing that biochar, palm fiber, and PAM enhanced runoff and had negative runoff reduction benefits for vegetation concrete slope. There was no significant difference in total runoff between the P, C, and F slopes under 60 mm·h$^{-1}$ rainfall intensity, but there was a significant difference in total runoff between the P and C slopes under 120 mm·h$^{-1}$ rainfall intensity ($p < 0.05$). The effect of PAM, biochar, and palm fiber in increasing runoff differed with the rainfall intensity. Further analysis of runoff reduction benefits revealed that the runoff reduction benefits of PAM, biochar, and palm fiber were −69.84~−1.97%, −68.82~−14.28%, and −63.70~−6.80%, respectively, with the corresponding average runoff reduction benefits being −32.72%, −38.82%, and −36.43%. The runoff reduction benefits of PAM, biochar, and palm fiber were reduced as the slope and rainfall intensity increased, indicating that the effect of PAM, biochar, and palm fiber on enhancing runoff declined as the slope and rainfall intensity increased.

**Table 4.** Runoff reduction benefit of the four treatments under different rainfall intensity and slope gradient.

| Runoff Intensity (mm·h⁻¹) | Slope Gradient (°) | Total Runoff (L) | | | | Runoff Reduction Benefit (%) | | |
|---|---|---|---|---|---|---|---|---|
| | | $R_{TCK}$ | $R_{TP}$ | $R_{TC}$ | $R_{TF}$ | $CR_P$ | $CR_C$ | $CR_F$ |
| 60 | 50 | 9.15 ± 0.12 b | 14.17 ± 0.21 a | 13.99 ± 0.18 a | 14.34 ± 0.25 a | −54.81% | −52.90% | −56.89% |
| | 60 | 9.44 ± 0.14 b | 16.03 ± 0.36 a | 15.94 ± 0.28 a | 15.45 ± 0.26 a | −69.84% | −68.82% | −63.70% |
| 120 | 50 | 19.96 ± 0.28 c | 20.81 ± 0.33 b | 22.81 ± 0.34 a | 21.32 ± 0.31 ab | −4.28% | −14.28% | −6.80% |
| | 60 | 22.26 ± 0.35 b | 22.70 ± 0.37 b | 26.55 ± 0.48 a | 26.34 ± 0.43 a | −1.97% | −19.28% | −18.34% |
| Average | | 15.20 | 18.43 | 19.82 | 19.36 | −32.72% | −38.82% | −36.43% |

Note: $R_{TCK}$, $R_{TP}$, $R_{TC}$, and $R_{TF}$ are the total runoff under the condition of control check (CK), 0.4% PAM (P), 4% biochar (C), and 0.4% palm fiber (F), respectively; $CR_{TCK}$, $CR_{TP}$, and $CR_{TC}$ are the runoff reduction benefit of the 0.4% PAM, 4% biochar, and 0.4% palm fiber, respectively. Different letters in the same row under the same rainfall intensity and slope gradient indicate significant differences between different treatments ($p < 0.05$). The same applies below.

Under different slope conditions, there were significant variances in total sediment, with a variation pattern of C > CK > F > P (Table 5). The average value of total sediment on the CK slope was 5.48, 0.61 and 2.09 times more than the average values on the P, C, and F slopes, demonstrating that PAM and palm fiber effectively reduced sediment yield, while biochar boosted sediment yield. The sediment reduction benefits of PAM, biochar, and palm fiber were 69.21~94.07%, −96.81~−50.35%, and 36.20~60.47%, respectively, with the corresponding average sediment reduction benefits of 84.79%, −66.72%, and 49.53%. PAM had a greater impact on sediment reduction than palm fiber. The sediment reduction benefits of the three materials showed different response laws to slope and rainfall intensity. The benefits of PAM sediment reduction were reduced as rainfall intensity and slope increased, as did the benefits of biochar increasing sediment yield. The sediment reduction benefits of palm fiber increased as rainfall intensity increased, decreased as slope increased under low rainfall intensity (60 mm·h⁻¹), and increased as slope increased under high rainfall intensity (120 mm·h⁻¹).

**Table 5.** Sediment reduction benefit of the four treatments under different rainfall intensity and slope gradient.

| Runoff Intensity (mm·h⁻¹) | Slope Gradient (°) | Total Sediment (g) | | | | Sediment Reduction Benefit (%) | | |
|---|---|---|---|---|---|---|---|---|
| | | $E_{TCK}$ | $E_{TP}$ | $E_{TC}$ | $E_{TF}$ | $CE_P$ | $CE_C$ | $CE_F$ |
| 60 | 50 | 59.32 ± 0.79 b | 3.52 ± 0.18 d | 116.75 ± 1.57 a | 32.02 ± 0.51 c | 94.07% | −96.81% | 46.02% |
| | 60 | 78.79 ± 0.84 b | 7.99 ± 0.22 d | 118.46 ± 1.68 a | 50.27 ± 0.60 c | 89.86% | −50.35% | 36.20% |
| 120 | 50 | 122.95 ± 1.63 b | 17.18 ± 0.31 d | 200.82 ± 2.55 a | 54.82 ± 0.69 c | 86.03% | −63.33% | 55.41% |
| | 60 | 151.43 ± 1.81 b | 46.62 ± 0.56 d | 236.79 ± 3.08 a | 59.86 ± 0.73 c | 69.21% | −56.37% | 60.47% |
| Average | | 103.12 | 18.83 | 168.21 | 49.24 | 84.79% | −66.72% | 49.53% |

Note: $E_{TCK}$, $E_{TP}$, $E_{TC}$, and $E_{TF}$ are the total sediment under the condition of control check (CK), 0.4% PAM (P), 4% biochar (C), and 0.4% palm fiber (F), respectively; $CE_{TCK}$, $CE_{TP}$, and $CE_{TC}$ are the runoff sediment benefit of the 0.4% PAM, 4% biochar, and 0.4% palm fiber, respectively. Different letters in the same row under the same rainfall intensity and slope gradient indicate significant differences between different treatments ($p < 0.05$).

## 4. Discussion

The initial runoff time, which is impacted by soil properties, ground slope, surface covering, rainfall intensity, and other factors, is an essential predictor of slope erosion [37,41–43]. The results of this study indicated that PAM, palm fiber, and biochar can significantly reduce the initial runoff time of vegetation concrete slope, with the degree of reduction differing. This is due to variations in the infiltration rate and porosity of vegetation concrete substrate caused by PAM, palm fiber, and biochar [32,44,45]. The application of PAM forms a layer of cementing material crust on the surface of the substrate, and the volume expansion after water absorption and dissolution forms a viscous gel to plug some soil pores, reducing the hydraulic conductivity of the substrate [46,47]. Studies have shown that although 4% biochar can effectively increase the porosity and permeability coefficient of vegetation concrete, it will reduce the dry density and make the surface of the substrate looser [7]. The pores of the soil surface will be sealed when

the fine particles formed by raindrop splashing are absorbed by biochar at the early stages of rainfall, which make the surface layer smoother and the resistance coefficient smaller, eventually leading to a reduction in the initial runoff time and affecting the runoff and erosion of the slope. The reciprocal coupling of soil–cement–palm fiber increases the mechanical characteristics of soil and lowers the infiltration rate of soil, which minimizes the initial runoff time of vegetation concrete with palm fiber [48]. The quantitative relationship between the initial runoff time and rainfall intensity and slope gradient is controversial. The reason might be due to the various soil types and surface cover conditions. However, it is generally believed that the initial runoff time decreases as rainfall intensity and slope gradient increase [49,50], which is supported by the findings of this study. This could be because rainfall on the slope increases with rainfall intensity, causing the time of infiltration excess runoff or saturation excess runoff on the slope to advance. Additionally, the time for rainfall to pass over the slope decreases as the slope gradient increases, accelerating the formation of runoff.

PAM, palm fiber, and biochar had minimal influence on the slope runoff process, with a tendency to grow first, then fluctuate and remain steady under each slope condition. The moisture of the substrate did not approach saturation during the early stages of rainfall, so some of the rainfall was absorbed by the substrate, and the runoff rate was relatively low [51,52]. The infiltration loss of the substrate was reduced as the water content of the surface substrate steadily grew, resulting in an increase in slope runoff, which then gradually stabilized. There is no obvious regularity between the runoff rate of the PAM, palm fiber, and biochar applied slopes under the combination of rainfall intensity and slope gradient, but it is generally greater than that of the control slope, and this difference is more pronounced when the rainfall intensity is low and the slope is steep. In contrast to the changing rules of the slope runoff process as rainfall duration increased, the erosion rate under each slope condition exhibited a tendency to fall initially, then fluctuate and stay steady. This is because the maintenance time was only 24 h, and the surface layer of the slope had not been completely cured. When it started to scour, the surface layer of soil broke apart due to the spattering effect of raindrops, and then the runoff took away a large number of soil particles. Simultaneously, the erosion rate followed the rule of C > CK > F > P. The erosion rate of a slope treated with biochar was reduced immediately and significantly, whereas the erosion rate of a slope treated with PAM, palm fiber, or control declined slowly or varied steadily. PAM flocculation [47] and palm fiber reinforcing [8] harden the surface layer of the substrate, but biochar lowers particle cohesion [7] and increases the quantity of floating soil, which is the primary cause of the aforesaid phenomenon. According to the observations, there was no visible rill on each slope during rainfall, in agreement with the results of Zhou et al. [13]. Because the microtopography of slope had not changed considerably, the runoff and erosion rates remained generally steady.

The total runoff of the vegetation concrete slope with PAM, palm fiber, and biochar used in this study was higher than the control slope, indicating that the runoff reduction benefits of PAM, biochar, and palm fiber were negative. The total sediment on each slope showed the pattern of C > CK > F > P-. PAM and palm fiber increase runoff while decreasing sediment yield, which are in general agreement with many experts [53,54]. PAM has a substantially greater sediment-reduction impact than palm fiber. The cement in the substrate has a high concentration of calcium and aluminum cations, which is advantageous for the mixing of soil particles with the anionic PAM [34,55]. Biochar has a positive influence on slope erosion while enhancing runoff and sediment yield. Contrary to prior findings that biochar has a positive effect on runoff and sediment reduction, the reason for this may be that the slope maintenance time is too short [56]. Slope gradient and rainfall intensity are important factors influencing rainfall runoff, and both have a considerable impact on net rainfall, slope runoff depth, runoff velocity, and slope shear stress, all of which affect soil infiltration, surface runoff, and soil loss [57,58]. The effect of biochar, palm fiber, and PAM on boosting slope runoff decreased as slope gradient and rainfall intensity increased, as did the sediment reduction effect of PAM. The sediment reduction impact of palm fiber exhibits a complicated rule as slope gradient and rainfall intensity change. The

slope gradient, rainfall intensity, and admixture application all have an interaction impact on the soil erosion at the early stage of vegetation concrete construction.

## 5. Conclusions

In this paper, we investigated the effects of PAM, palm fiber, and biochar on the process of runoff and sediment yield on the slope under different rainfall intensities and slope gradients at the early stage of vegetation concrete construction by the simulated rainfall experiment. The application of 0.4% PAM, 0.4% palm fiber, and 4% biochar significantly reduced the initial runoff time of the vegetation concrete slope, with 0.4% PAM and 0.4% palm fiber being more effective than 4% biochar. Meanwhile, 0.4% PAM, 0.4% palm fiber, and 4% biochar all increased runoff rates, with PAM and palm fiber greatly reducing erosion rates. The linear equation $S_L = AR_L + B$ could better express the relationship between cumulative sediment and cumulative runoff, with cumulative sediment increasing as cumulative runoff increases.

Furthermore, the total runoff of the vegetation concrete slope with 0.4% PAM, 4% biochar, and 0.4% palm fiber was greater than that of the control slope, and the runoff reduction benefits were as follows: $-69.84\sim-1.97\%$, $-68.82\sim-14.28\%$, and $-63.70\sim-6.80\%$. The total sediment of each slope showed a changing rule of 4% biochar > control > 0.4% palm fiber > 0.4% PAM. The sediment reduction benefits of 0.4% PAM, 4% biochar, and 0.4% palm fiber were $69.21\sim94.07\%$, $-96.81\sim-50.35\%$, and $36.20\sim60.47\%$, respectively. Both 0.4% PAM and 0.4% palm fiber have evident sediment reduction benefits, with 0.4% PAM having a substantially greater sediment reduction effect than 0.4% palm fiber.

Because the results of this greenhouse simulation might be different than the results that one might see on real constructed slopes, more research needs to be performed to determine the effect of the application of PAM treatment and palm fiber treatment and to verify the effect on plant growth. Overall, the results of this study could promote polyacrylamide and palm fiber application in the ecological restoration of high and steep slopes in areas with heavy rainfall.

**Author Contributions:** L.X.: paper framework, conceiving and planning the experiments, writing and providing funding; B.Z.: sample collection, data processing, writing, providing funding; T.L.: sample collection, writing; Y.X.: revising the manuscript, S.G.: sample collection, data processing; W.X.: data processing, providing funding; D.X.: conceiving and planning the experiments, re-writing, providing funding. All authors have read and agreed to the published version of the manuscript.

**Funding:** This study was financially supported by the National Natural Science Foundation of China (Grant no. 52200230), the National Natural Science Foundation of China (Grant no. 51979147), the Major Science and Technology Projects of Inner Mongolia Autonomous Region (2021ZD0007-03), and Research Fund for Doctoral Dissertation of China Three Gorges University (Grant no. 2020BSPY002).

**Data Availability Statement:** Data will be shared upon request.

**Conflicts of Interest:** The authors declare no conflict of interest.

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
