# Peer review of "Effects of Polyacrylamide, Biochar, and Palm Fiber on Soil Erosion at the Early Stage of Vegetation Concrete Slope Construction"

_sustainability, doi:10.3390/su15075744_

Round 1
Reviewer 1 Report
After reviewing this manuscript, I found that the authors have done an interesting research. I suggest that this manuscript be published after minor revision. The specific modification suggestions are as follows.
1. Line 34~35: The definition of CBS is not described clearly enough, and it is suggested to re-describe it in one or two sentences.
2. Line 122: Is it a writing error? There are three rainfall intensities here and only two in the following.Please check carefully.
3. “Different letters in the same row under the same rainfall intensity and slope gradient indicate significant differences between different treatments (P<0.05). The same below.” Please explain the reason for the selection of the value. (Line 241~242)
4. The addition of the three materials also changes the roughness of the slope surface, which in turn affects the initial runoff time, suggesting that the authors could enrich the relevant discussion from this perspective.(Line 284~297)
5. “flow reduction benefits”is the “runoff reduction benefits”? (Line 324). Professional terms should be used consistently throughout the paper. Please check carefully.
6.The conclusion is not concise enough and should be re-condensed.
Reviewer 2 Report
The manuscript on "Effect of polyacrylamide, biochar, and palm fiber on soil erosion at 1 the early stage of vegetation concrete slope construction" is an useful contribution by the authors and it is nicely written. I have following queries/suggestions on the manuscript:
1) How the proportions of admixtures are fixed? Is it random in nature or any reference for the same?
2) Line 163, correct the spelling of "slope"
3) In all figures it is mentioned as "four slope conditions". But it is four combinations with slope and rainfall intensity. This needs to be corrected which otherwise misleads.
4) Lines 176-177, "With the increase in rainfall intensity and slope, the initial runoff time of different slopes also decreased". What could be the reason? Please justify.
5) Lines 192-194, ".......with the impact being more pronounced when the rainfall intensity is low." Is this because of imperviousness of the top layer?
6) Why there is heavy fluctuations both in runoff and erosion rates? Is there any experimental errors?
7) Lines 229-230, "With the increasing rainfall intensity and slope, the A value of each slope essentially increased". However, for CK A has decreased with rainfall intensity even though it is increased with slope?. Generalization of statement is not possible.
8) What are 'a b c d' in Tables 3 and 4?
9) Line 287, "..... making the surface layer smoother, the resistance coefficient smaller, " Will this not affect the surface flow? Please highlight this.
10) Did the author observe the rill formation on the slopes with the treated surfaces?
11) Is the treatment will not affect the soil and vegetation conditions?
Reviewer 3 Report
1. It is better to show a picture of the rainfall sprinkler system.
2. It should explain how to make and mix each of the different samples.
3. It is good to explain the cost per square meter of different layers and to compare each of the samples in this regard.
4. The reasons for the obtained results should be explained and the results should not be written in reports.
